# VSL-Skin: Voxel-Level Variable Stiffness for Predictable Deformation and Robust Manipulation Control

*Abstract*—Soft robotic manipulators offer compliance for safe interaction but suffer from unpredictable deformation under contact and load, making robust manipulation control difficult. We present the Variable Stiffness Lattice Skin (VSL-Skin), a conformal robotic skin comprising individually addressable phase-change voxels arranged in a triangular lattice. By selectively melting or solidifying voxels via Joule heating, the skin programs discrete virtual joints and canonical deformation modes at centimeter-scale resolution across axial, shear, bending, and torsional modes, with nearly two orders of magnitude stiffness modulation demonstrated. The key insight for manipulation robustness is that voxel-level stiffness control converts the open-ended deformation space of a compliant manipulator into a discrete, plannable set of known deformation modes. Under a given activation pattern, deformation is mechanically constrained to a preferred mode with predictable stiffness and range of motion, restoring determinism to an otherwise uncertain system. We further identify that the carbon-black–silicone heater matrix exhibits piezoresistive behavior, presenting a pathway toward distributed proprioceptive sensing through the same electrode network, and that the programmable voxel lattice can serve as a physical AI substrate for embodied manipulation intelligence.

*Index Terms*—variable stiffness, robust manipulation, soft robotics, virtual joints, morphological computation, embodied AI

> **Supplementary Video:** System demonstrations are available at: [link].

This work is submitted into ICRA 2026

## I. INTRODUCTION

Robust manipulation requires predictable, controllable mechanical behavior under uncertainty in contact, load, and environment. Soft robots offer compliance and safe interaction, but their deformation under external forces is difficult to predict or constrain, making planning and control challenging [1], [2]. Rigid robots resolve this by constraining deformation mechanically, but sacrifice the compliance needed for safe and adaptive contact. Variable-stiffness systems attempt to bridge this gap, but most operate at patch or segment scale and are embedded into bespoke morphologies, limiting the spatial resolution of stiffness control and the ability to impose specific deformation patterns [3], [4].

We present VSL-Skin, a platform-agnostic robotic skin that achieves individually addressable voxel-level stiffness control through selective phase-change actuation of a triangular lattice. Unlike variable-stiffness mechanisms embedded into custom designs, VSL-Skin is a skin: it wraps onto existing soft manipulators and structures and endows them with centimeter-scale

morphological programmability without requiring redesign of the underlying body. A soft continuum arm that previously exhibited uncontrolled deformation gains discrete, predictable virtual joints simply by having the skin applied to it.

The central contribution to manipulation robustness is a shift in how deformation uncertainty is handled. Rather than tolerating unpredictable compliance and designing controllers that compensate for it, VSL-Skin programs preferred deformation modes directly into the mechanical substrate. Under a given activation pattern, deformation is mechanically biased toward a known mode with predictable stiffness and range of motion. This converts the continuous, high-dimensional deformation space of a soft manipulator into a discrete set of canonical modes that can be planned over, scheduled, and composed. The controller then manages a discrete activation space rather than an uncertain continuous body.

## II. SYSTEM OVERVIEW

### A. Design and Fabrication

VSL-Skin comprises equilateral triangular voxels (18 mm side length) arranged in a conformal lattice. Each voxel houses a Field's metal slug ($T_m \approx 62°C$) thermally coupled to a resistive heater fabricated from carbon-black–loaded silicone (Super P, 11 wt% in Ecoflex 30) with embedded copper electrodes, enabling independent switching between rigid and compliant states via localized phase change. A row-column addressing scheme enables per-voxel control through standard PWM harnesses. The triangular lattice was selected through comparative FEA across seven candidate geometries under equal-mass constraints, achieving the highest normalized stiffness across all mechanical modes. Per-voxel thermal calibration stores duty-cycle-to-temperature maps and thermal time constants, correcting for fabrication variability and enabling reliable autonomous group activations.

Because each voxel is self-contained, the skin can be trimmed to match any surface geometry, re-terminated at the cut edge, and deployed immediately. Adapting the skin to a new manipulator geometry requires no redesign of the voxel itself, only a trim and re-termination step, substantially lowering the integration barrier.

### B. Stiffness Programmability

Selective voxel activation achieves nearly two orders of magnitude stiffness modulation across mechanical modes, with axial stiffness ranging from 15 to 1200 N/mm, shear from 45 to

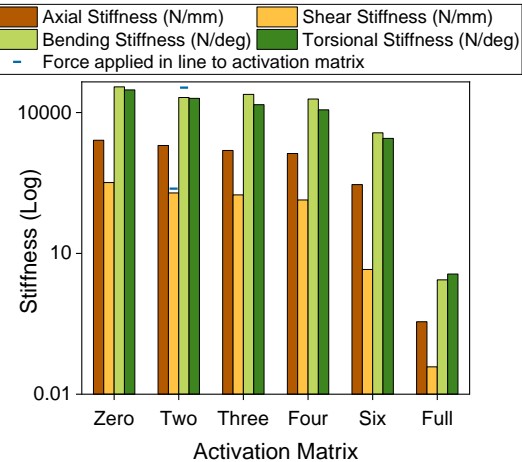

Fig. 1. Programmable stiffness range (log scale) across axial, shear, bending, and torsional modes. Discrete activation sets from zero to twelve active voxels demonstrate stepwise modulation without host structure reconfiguration.

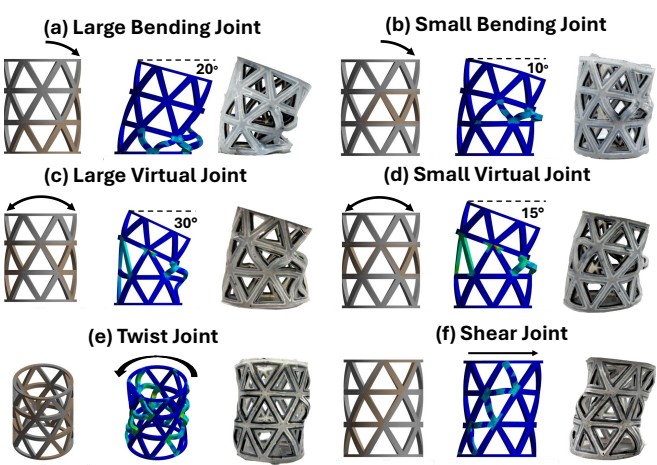

Fig. 2. Six canonical joint configurations (a–f). Each panel shows the activation matrix (brown = activated, gray = deactivated), simulated elastic strain field, and experimental deformed shape. Strain localizes to activated voxels in all cases, enabling precise placement of compliant zones.

850 N/mm, and bending from $8 \times 10^2$ to $3 \times 10^4$ N/deg (Fig. 1). Six canonical virtual joint configurations are demonstrated through distinct activation patterns, including unilateral bending joints, bilateral virtual hinges, localized twist joints, and lateral shear joints (Fig. 2). In all cases, strain localizes to the activated region while non-activated neighbors remain stretch-dominated, producing low cross-talk between co-located joint patterns. The system also achieves up to 30% axial compression through simultaneous circumferential activation, with re-solidification restoring full stiffness in the shortened pose.

Because actuation is per-voxel, virtual joints degrade gracefully under single-voxel failure and can be rerouted by updating activation patterns. Re-melting followed by re-solidification restores the mechanical state after overload, eliminating fatigue accumulation and enabling on-demand self-repair.

## III. ROBUST MANIPULATION THROUGH PROGRAMMABLE MORPHOLOGY

### A. Reducing Deformation Uncertainty via Canonical Modes

The fundamental challenge in soft manipulator control is that deformation under contact is not uniquely determined by the actuator input: contact geometry, load distribution, and material nonlinearity all contribute to an uncertain deformation outcome. VSL-Skin addresses this by constraining the deformation space mechanically before the controller acts. Under a given activation pattern, the stiffness field is spatially fixed, and deformation is biased toward the mechanically preferred mode encoded by that pattern. The controller then operates on a substrate whose response is known in advance.

This is not merely convenience. Because strain localizes to activated voxels by construction, the deformation mode is determined by the activation pattern rather than by contact conditions. A gripper fitted with VSL-Skin and programmed with a bilateral-hinge pattern will deform at the hinge location regardless of where along the finger contact is made, as long as the contact force is within the compliant zone's load range. This provides a form of mechanical robustness to contact uncertainty that does not depend on contact sensing or real-time compensation.

Quantitatively, widening the bilateral-hinge activation band shifts rotation from $15°$ to $30°$ with a corresponding stiffness reduction, and the shear pattern increases shear stiffness by 24% and bending stiffness by 62% over the all-compliant baseline. These are reproducible, predictable responses that can be looked up from a calibrated activation-to-response map rather than computed online.

### B. Planning over Discrete Deformation Programs

Rather than planning over the continuous configuration space of a deformable body, a planner operating on VSL-Skin selects from a discrete library of activation patterns, each corresponding to a known joint type, location, stiffness, and range of motion. The planning problem reduces to selecting and sequencing activation patterns that achieve the desired manipulation trajectory, with each step producing a mechanically enforced constraint on deformation.

By combining patterns into time-phased voxel commands, complex manipulation sequences can be composed from canonical primitives. Since patterns are expressed in row-column coordinates, the same joint library transfers across different host morphologies without modification, meaning plans developed for one manipulator geometry can be reused on another with the same skin. This reduces the need for per-platform replanning and supports generalization across manipulation contexts.

The six demonstrated joint types provide a vocabulary of manipulation primitives with distinct mechanical signatures: unilateral bending joints for curvature fields around contact surfaces, bilateral hinges for symmetric reversible bending, twist joints for end-effector orientation control, and shear joints for lateral compliance during insertion and seating tasks.

Composing these covers a broad range of manipulation sub-tasks without requiring continuous body control.

### C. Compliance and Variable Stiffness as a Control Resource

Compliance in soft robotic manipulation is typically treated as a disturbance to be compensated. VSL-Skin reframes it as a programmable control resource. Anisotropic stiffness patterning allows the skin to selectively attenuate specific load components before they propagate to the controller or to the manipulated object. A pattern that increases shear stiffness while maintaining low bending stiffness allows lateral forces from the environment to be absorbed passively while preserving active bending control. This mechanical pre-conditioning reduces the number of disturbances the control algorithm must handle in real time.

Re-solidification after a manipulation step locks the current configuration at high stiffness, providing pose retention without continuous actuation. This is directly relevant to robust manipulation: a grasped object can be held rigidly while the arm repositions, with compliance re-enabled at a subsequent grasp adjustment step, without requiring active force control throughout.

## IV. Toward Sensing and Embodied AI

### A. Integrated Proprioception Through the Heater Matrix

Carbon-black–silicone composites are well established as piezoresistive strain sensors, with resistance varying repeatably under deformation above the percolation threshold [5], [6]. The VSL-Skin heater matrix uses Super P carbon black in Ecoflex at 11 wt%, well above this threshold. Because the heater shares the same electrode network already routed to each voxel, the actuation interface presents a direct path toward distributed proprioceptive sensing with no additional hardware. The actuation and sensing interfaces would be physically the same object, closing the information loop between programmed morphology and measured deformation. Integrating multiplexed readout into the existing per-voxel calibration framework is the immediate next step.

Proprioceptive sensing at voxel resolution would provide direct feedback on whether the programmed deformation mode has been achieved and whether contact forces are within the expected range for the active pattern. This supports closed-loop verification of manipulation state without requiring external vision, which is a significant robustness advantage in occluded or cluttered environments.

### B. Physical AI for Adaptive Manipulation

Framing each voxel as a node in a physical AI network suggests a further direction for manipulation robustness. Stiffness state governs how mechanical signals propagate through the lattice, so an activation pattern defines a physical transfer function between applied loads and nodal displacements readable through piezoresistive channels. This is consistent with physical reservoir computing [7], where substrate dynamics project inputs into a high-dimensional mechanical state space from which task-relevant information is extracted by a lightweight readout layer. Unlike fixed-topology reservoirs, the network here is reconfigurable on demand, enabling the substrate to be tuned to different manipulation tasks through the activation schedule alone. This pathway toward embodied AI, where the mechanical substrate itself participates in computation and decision-making, complements the discrete planning layer described above and reduces the demand on centralized control.

## V. Discussion and Conclusion

VSL-Skin demonstrates centimeter-scale, voxel-level stiffness programmability across axial, shear, bending, and torsional modes, with virtual joint generation, 30% axial compression, autonomous self-repair, and platform-agnostic retrofit deployment validated experimentally. The core contribution to manipulation robustness is mechanical: by programming preferred deformation modes into the substrate, the skin converts an uncertain compliant body into a discrete-state system that can be planned over and controlled with deterministic expectations. Compliance is no longer a source of uncertainty to be compensated but a programmable resource to be scheduled.

The piezoresistive pathway through the heater matrix points toward closed-loop proprioceptive verification of manipulation state, and the physical AI framing toward adaptive on-body computation. Together, these suggest that programmable morphology, sensing, and control are most effectively treated as co-designed layers that collectively enable manipulation robustness, rather than as separate concerns addressed in isolation.

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
