# OpenReview forum: "VSL-Skin: Voxel-Level Variable Stiffness for Predictable Deformation and Robust Manipulation Control"
_IEEE.org/ICRA/2026/Workshop/Manipulation_Robustness — ICRA 2026_

### Official Review · Reviewer_TfCY · 2026-05-06
**Interesting work on variable-stiffness robotic skin but more experiments on grasp/manipulation expected**

**Rating:** 6
**Confidence:** 3

**Review:**

Reviewer 1:
Strengths:
This paper proposes VSL-Skin, a voxel-level variable-stiffness robotic skin that enables programmable deformation modes for soft manipulators. The hardware concept is creative and well aligned with manipulation robustness, particularly the idea of discretizing a continuous deformation space into a set of canonical, controllable modes. The skin-like, retrofittable design is practically appealing, and both the paper and supplementary video show convincing mechanical behavior, including stiffness modulation, localized virtual joints, and integration onto a soft robotic structure. The self-repair capability demonstrated in the video is also a promising feature, even though it is currently qualitative.

Weaknesses:
The main limitation is that the paper does not provide direct validation of manipulation robustness. Both the experiments and video focus on mechanism-level demonstrations (deformation modes, joint activation, manual loading), without task-level evaluation such as grasping under uncertainty, disturbance rejection, or comparison with baselines (e.g., skinned vs. un-skinned systems). The self-repair behavior is shown qualitatively, but there is no quantitative evidence that performance (e.g., stiffness or function) is fully restored. Additionally, some deformation experiments appear to be conducted without external load, making it unclear how the system behaves under realistic interaction forces, and the implications of localized strain (e.g., stress concentration or durability) are not discussed. Finally, sections on proprioception and “physical AI” are speculative and would benefit from clearer separation from validated contributions.

Overall comments:
This is a creative and relevant hardware contribution with a compelling conceptual framing of programmable morphology as a control interface. While the current evaluation remains limited to mechanism-level demonstrations, the idea is promising and suitable for discussion in a workshop setting. I would encourage the authors to focus the contribution more clearly and include at least a simple task-level experiment or baseline comparison in future iterations.

///////////////////
Reviewer 2:
The paper shows the concept of VSL-Skin, a lattice, which can change its stiffness for each voxel, by phase change of the fields metal inside the lattice. This is an interesting idea, also the use of heaters as sensors. But the integration of  this feature and the cabling for heating is missing. Also it is unclear if the lattice can go back fully to its start position and how the stiffness of the internal soft structure compared to the skin  must be chosen.  Furthermore a rough phase change timing would be interesting.

///////////////////
Reviewer 3:
The core idea of reframing compliance as a schedulable control resource, encoded directly into voxel activation patterns, is a genuine and well-motivated contribution to hardware-control co-design. The mechanical characterization backs this up well, with stiffness modulation across four modes and six canonical joint configurations validated through both simulation and experiment.That said, the robustness claims go well beyond what the experiments actually support. Everything is tested quasi-statically under no-load conditions, and without any grasping, insertion, or contact-rich tasks, it is hard to take the robustness argument at face value. Real manipulation experiments would make the case much more convincing.The physical AI section reads more like speculation than a concrete contribution. It would be interesting to see even a simple simulation or experiment connecting activation patterns to actual task performance. Some quantitative comparison to prior variable-stiffness approaches like layer jamming or SMA-based skins would also help ground the paper's claims and make it clearer how much of an advance this work actually represents.

---

### Decision · Program_Chairs · 2026-05-21

Accept